# VE-Cadherin in Cancer-Associated Angiogenesis: A Deceptive Strategy of Blood Vessel Formation

**DOI:** 10.3390/ijms24119343

**Published:** 2023-05-26

**Authors:** Daniel Delgado-Bellido, F. J. Oliver, María Victoria Vargas Padilla, Laura Lobo-Selma, Antonio Chacón-Barrado, Juan Díaz-Martin, Enrique de Álava

**Affiliations:** 1Instituto de Parasitología y Biomedicina López Neyra, CSIC, 18016 Granada, Spain; ddelgado-ibis@us.es (D.D.-B.);; 2Instituto de Salud Carlos III, CIBERONC, 28220 Madrid, Spain; 3Instituto de Biomedicina de Sevilla, Hospital Virgen del Rocío, 41013 Seville, Spain; 4Department of Normal and Pathological Cytology and Histology, School of Medicine, University of Seville, 41009 Seville, Spain

**Keywords:** vasculogenic mimicry, VE-cadherin, angiogenesis

## Abstract

Tumor growth depends on the vascular system, either through the expansion of blood vessels or novel adaptation by tumor cells. One of these novel pathways is vasculogenic mimicry (VM), which is defined as a tumor-provided vascular system apart from endothelial cell-lined vessels, and its origin is partly unknown. It involves highly aggressive tumor cells expressing endothelial cell markers that line the tumor irrigation. VM has been correlated with high tumor grade, cancer cell invasion, cancer cell metastasis, and reduced survival of cancer patients. In this review, we summarize the most relevant studies in the field of angiogenesis and cover the various aspects and functionality of aberrant angiogenesis by tumor cells. We also discuss the intracellular signaling mechanisms involved in the abnormal presence of VE-cadherin (CDH5) and its role in VM formation. Finally, we present the implications for the paradigm of tumor angiogenesis and how targeted therapy and individualized studies can be applied in scientific analysis and clinical settings.

## 1. A Glance at Angiogenesis Past

Vessel formation by tumor cells represents multiple functionalities by tumor cells that recently have more attention from the scientific community. The vision of new vessels or neovascularization was first described in 1787 regarding the enhancing tissues of animals [1]. Angiogenesis was coined in the 1900s but was not used in tumors until the 1960s [2]. Several types of angiogenesis have been reported in the cancer context: sprouting angiogenesis, intussusception microvascular growth [3], glomeruloid microvascular proliferation, vessel co-option, and vasculogenic mimicry (VM) [4]. Recent studies show that only a few tumors can grow essentially non-angiogenic, even under hypoxic conditions, and other tumors have a mixture of angiogenic and non-angiogenic areas [5]. VM is concerned with the faculty of cancer cells to coordinate themselves into vascular-like structures to obtain nutrients and oxygen solo of normal blood vessels or angiogenesis [6].

Vessels can expand in various aspects. While vasculogenesis refers to the organization of blood vessels by endothelial cell progenitor cells, angiogenesis refers to the formation of sprouts and their ensuing stabilization by mural cells, and arteriogenesis is defined as collateral growth denoting the growth of blood vessels, expansive of the pre-existing vessels, finally forming collateral bridges between networks of arteries. When the development of the vessels is deregulated, it can affect our health and produce various pathologies [7]. This dysregulation may be due to inflammation of the endothelial barrier or joint leukocyte infiltration, as well as other pathologies, such as cancer and hypoxia, occurring within a microenvironment well-orchestrated by tumor cells. Other pathologies implicated include psoriasis, arthritis, obesity, asthma, arteriosclerosis, and infectious diseases, and the list continues to grow [7]. Until now, traditionally, the treatment of vessel-blocking strategies has been using anti-angiogenic drugs. These try to inhibit the formation of new vessels and shatter existing vessels produced by cancer cells [8,9].

In the 1990s, Pezzella et al. [10,11] showed that several lung tumors co-opt pre-existing vessels rather than induce angiogenesis. The effectiveness of anti-angiogenic agents in patients may be significantly impeded by this procedure. However, several anti-angiogenic agents regulate VEGF (vascular endothelial growth factor) signaling and have been implemented in clinical settings, including bevacizumab (an antibody directed to VEGF-A), ramucirumab (an antibody directed to VEGFR2), aflibercept (a VEGF trap), and several VEGFR tyrosine kinase inhibitors (such as nintedanib, pazopanib, regorafenib, sorafenib, sunitinib, and vatalanib). These agents have been used to treat patients with high-stage lung, kidney, gastrointestinal, brain, liver, breast, thyroid, pancreas (pancreatic neuroendocrine tumor), cervix, and ovary [12] (Figure 1). Nevertheless, although sustained angiogenesis is a prerequisite for tumor progression, the anticipated therapeutic success of anti-angiogenic therapy has not been fully realized [13]. Initially, despite the assumed universal dependency, anti-angiogenesis therapy agents are not regularly active in all types of tumors. Reproducible clinical trials have shown statistically significant improvements in disease-free survival (DFS) and overall survival (OS) for certain cancers, including high-stage renal cell carcinoma (RCC), hepatocellular carcinoma (HCC), and colorectal carcinoma (CRC) [14,15,16,17,18,19]. However, for pancreatic adenocarcinoma, prostate cancer, breast cancer, or melanoma, no significant improvements in OS have been demonstrated with anti-angiogenic agents, either alone or in combination with other treatments [20,21,22,23,24]. One possible cause of the failure of anti-angiogenic therapy in some cases is the ability of aggressive and dedifferentiated cancer cells to undergo non-productive angiogenesis under hypoxic conditions. This mechanism enables cancer cells to survive and proliferate in hypoxic environments despite the presence of anti-angiogenic agents, contributing to the failure of anti-angiogenic therapy. Essentially, these cells employ non-productive angiogenesis to sustain their growth and can resurface when conditions become favorable again. In the following sections, we describe the different models of known production of angiogenesis in cancer, with a significant focus on the mechanism called VM, thus representing in Figure 1 the most important scientific advances in this field [2].

## 2. Facts

There is a correlation between VM and high tumor grade, cancer cell invasion, cancer cell metastasis, and reduced survival of cancer patients;VE-cadherin, in conjunction with focal adhesion kinase (FAK) activity and tethered β-catenin, serves as a regulator of the tumor microenvironment, promoting the formation of VM;Anti-angiogenic therapy fails in several types of cancer in overall survival;FAK plays a pivotal role in the dynamics of cell permeabilization in the metastatic tumor endothelium.

## 3. Mechanism of Vascularization in Cancer: Endothelial Sprouting

Tumors commonly promote their vascularization through the induction of new capillary sprouting from pre-existing host tissue capillaries. The process was first reported in the 1970s by Ausprunk and Folkman [25], who proposed the following steps for tumor-induced capillary sprouting:Angiogenic stimuli can cause the local degradation of the basement membrane on the side of the post-capillary venule that is dilated and located closer to the tumor. This can weaken the inter-endothelial contacts, and the endothelial cells (ECs) may migrate into the surrounding connective tissue;A solid bead is then formed by the ECs that accomplish each other in a bipolar manner;The formation of blood vessels occurs through the curvature of one or more ECs in conjunction with forming a new basement membrane and recruiting the pericytic or mural cells.

This model has significant drawbacks, such as its inability to identify the origin of the initial and ongoing stimulus for lumen formation. It also assumes that dedifferentiation and redifferentiation occur during the same process and time, as evidenced by the loss and recovery of the luminal-basal polarity of ECs. Furthermore, although the stimulus required for lumen production has been reported to come from the developing basement membrane, this model suggests that basement membrane deposition occurs after lumen formation [26].

In the early 1990s, a different model of endothelial sprouting was described [25]. This model proposes a three-stage sequence to explain the ultrastructural changes that occur during tumor-induced endothelial sprouting:Structural alteration of the basement membrane due to a loss of electron density in almost the entire area of the dilated “original vessel”. However, immunohistochemical techniques can still locate basement membrane components such as laminin and collagen IV. The impaired basement membrane’s partially regulated degradation occurs only at sites where endothelial cell processes, connected by gap junctions, project into the surrounding connective tissue;The migration of ECs, which maintain their basal-luminal polarity and form a slit-like lumen in parallel with the lumen of the main vessel, takes place continuously and is stamped by intact inter-endothelial junctions. The low electron density basement membrane is continuously deposited by polarized ECs, while only the tip of the growing capillary is free of basement membrane material;Pericytes [27] from the mother vessel relocate along the basement membrane of the capillary sprout, resulting in complete coverage of the new vessel. Similarly, the appearance of an electron-dense basement membrane can be observed around the mature capillary buds. In contrast to the previous model, this model suggests that the loss of endothelial cell polarity is not a necessary stimulus for the induction of lumen formation.

The molecular mechanisms underlying the sprouting process have been extensively studied and reviewed [28]. Initially, the vessels dilate and become weaker in response to the vascular permeability factor (VPF/VEGF) [29], but this is resolved through regulation by nitric oxide, expansion of fenestrations and vesicular–vacuolar organelles, and redistribution of CD31/PECAM-1 and VE-cadherin. The transition of the basement membrane is likely mediated by matrix metalloproteinases (MMP), gelatinases, and the urokinase plasminogen activator system and may play a role in initiating endothelial cell proliferation and migration. Shedding of pericytes and matrix disintegration are mediated by Ang-2, a mediator of Tie-2 signaling [30]. Various molecules stimulate endothelial proliferation and migration, including transforming growth factor (TGF-1), tumor necrosis factor (TNF-α), members of the chemokine system, fibroblast growth factor, and platelet-derived growth factor (PDGF) [31]. Integrins are considered the most important adhesion receptors for the migratory capacity of ECs [32]. Sprouting is a complex process tightly regulated by a delicate balance between pro- and anti-angiogenic factors. Angiogenic cytokines play a crucial role in promoting the proliferation, migration, and lumen formation of blood vessels, while inhibitor cytokines act to modulate these steps and regulate angiogenesis. The specific combination of pro-angiogenic and anti-angiogenic inhibitory cytokines used by different tumor types varies, depending on the tumor’s angiogenic activity [33].

## 4. Mechanism of Vascularization in Cancer: Vessel Co-Option

Tumors commonly develop and expand within pre-existing, well-vascularized tissues, or they may metastasize to such tissues. Additionally, tumors frequently emerge in tissues affected by chronic inflammatory processes, wherein the inflammatory infiltrate and fibrosis contribute to an ischemic state. Their growth depends not only on expansion in their original location, which is more typical of slow-growing benign tumors, but also on invasion and expansion away from the primary tumor site. Cancer cells need to come into close contact with the surface of blood vessels, either closely at initiation or during an expansion [34], to continue growing. Malignant cells can primarily associate with and preferentially grow along pre-existing microvessels. Until recently, little attention had been given to the role of the host’s vasculature and the use of pre-existing vessels in the process of tumor vascularization.

Although it was proposed in 1987 that tumors acquired their vasculature by incorporating capillaries from the host tissue, the first study suggesting the existence of vessel co-opting was not published until 1999 by Holash et al. [35]. In developing this model, Holash and colleagues found that co-optation is limited to the early phases of tumor growth. However, more recent evidence suggests that co-optation of pre-existing blood vessels might persist throughout the growth period of the primary or metastatic tumor. For example, in cutaneous melanoma, it was shown that there are no signs of directed vessel growth during tumor growth; instead, these tumors appear to grow by co-opting the massive vascular plexus present in the peritumoral connective tissue [36]. Similarly, in non-small cell lung cancer, a putatively non-angiogenic growth pattern was observed [11], where tumor cells occupied the alveoli, entrapping but not destroying the co-opted alveolar capillaries.

Different growth patterns were observed in liver metastases from human colorectal carcinomas, depending on the degree of cell differentiation. In the replacement growth type, the architecture of the liver was preserved, and the ECs of the sinusoids showed low mitotic activity. However, the pushing and desmoplastic types of tumors destroyed the architecture of the liver. Overall, tumors can use pre-existing vessels to facilitate their growth and invasion, and different types of tumors exhibit different growth patterns that reflect their level of vascular co-optation.

Vessel co-optation is a process in which cancer cells use pre-existing blood vessels in non-malignant tissue to invade and colonize a target area without stimulating the growth of new vessels through angiogenesis. This is achieved by cancer cells penetrating the outer surface of pre-existing vessels and occupying the tissue space between them, resulting in the integration of these vessels into the tumor. Vessel co-optation allows cancer cells to use the host’s vessels to meet their metabolic needs, potentially explaining the limitations of anti-angiogenic therapies, including their frequent failure due to intrinsic or acquired resistance [4,37,38,39,40,41]. Vessel co-option plays a crucial role in the metastatic process, which involves the spread of cancer cells from the primary tumor to distant organs. Cancer cells that utilize co-opted vessels can enter the circulation and reach distant organs without triggering an angiogenic response. Co-opted vessels can also provide a supportive microenvironment for cancer cells during their extravasation into the target tissue and their survival at the secondary site. Additionally, the survival of cancer cells during latency, a period during which they are present in a dormant state in the secondary site without growing, can also be facilitated by co-opted vessels. Thus, vessel co-option provides a mechanism for cancer cells to adapt to different environments, evade therapy, and establish secondary tumors [42].

The histopathological analysis is the standard method used to identify vessel co-option in cancer tissues. Several features can indicate the recruitment of pre-existing blood vessels, including cancer cells surrounding pre-existing vessels without evidence of angiogenic sprouting and the presence of capillary-like structures within the tumor. Vessel co-option has been observed in a variety of cancer types in different tissues, including the lung, liver, brain, skin, and lymph nodes. In some cases, vessel co-option occurs without concurrent evidence of angiogenesis [43]. Recently, several seminal articles [44,45] have focused on single-cell profiling in the context of vessel co-option formation from cancer cells. This new approach provides a longitudinal view with precision and individualized assessment of the tumor cells involved in aberrant angiogenesis formation.

## 5. Mechanism of Vascularization in Cancer: Vasculogenic Mimicry

Cancer cells necessitate blood vessels for growth and are the gateway to oxygen and nutrients. Current anti-angiogenic therapies are displayed to target vascular ECs capable of forming blood vessels de novo [46]. Although various pre-clinical models have acknowledged the efficacy of angiogenesis inhibitors in restraining tumor growth, it is worth noting that only the hindrance of growth has proven effective in clinical practice [47]. Part of the difficulty in achieving a comprehensive understanding of tumor vasculature stems from the complexity of the variables and alterations that act upon the tumor environment. As a result, many studies in cancer pathology have focused on describing the elevated plasticity, or embryonic-like cells, associated with aggressive cancer. This plasticity and embryonic phenotype may explain the ability of aggressive tumor cells to mimic the functions of normal ECs and participate in neovascularization processes, including the formation of pseudo-ducts that conduct plasma through a connected vascular network [48,49].

In 1999, Maniotis [50] et al. presented the term VM to explain the unique ability of extremely aggressive PAS (periodic acid–Schiff) positive and CD31 negative uveal melanoma tumor cells to form tubular structures and three-dimensionally patterned networks in cellular media, which “mimic” primitive vasculogenic networks formed by dedifferentiation of tumor cells.

The term “vascular mimicry” describes the formation of channels by non-ECs, which have the ability to conduct fluids such as plasma and red blood cells. This process is categorized into two types: (1) mimicry, where the channels are not actual blood vessels but simply mimic their function, and (2) vasculogenic, where the channels do not form from pre-existing vessels but still dispense plasma and may contain blood cells [51]. VM provides tumors with an angiogenesis-independent perfusion pathway without the participation of ECs. Various interpretations of VM have been proposed, stemming from different analyses of the original observations. A basic explanation of VM is the identification of PAS-stained vascular networks within tumors. Additional characteristics of VM include cancer cells lining the channels or gaps in the network, as well as the presence of erythrocytes or blood lakes. Some researchers have suggested that VM may be attributed to tumor cells expressing specific genes typically found in ECs [52]. It is plausible that any combination of these contexts could explain the characteristics of the VM. The term “vascular mimicry” has also been used simultaneously with VM. Nevertheless, vascular mimicry has immense implications as it includes other cell-associated phenotypes, such as lymphocytes and macrophages [53,54]. VM has been inspected in osteosarcoma, breast cancer, colon cancer, hepatocellular carcinoma, small-cell lung cancer, glioma, ovarian cancer, prostate cancer, head and neck cancer, Ewing sarcoma, cutaneous melanoma, and uveal melanoma [55]. The existence of VM in tumors is indicative of aggressiveness and clinically corresponds to a 50% increased risk of death from metastasis [56]. Kaplan Meier survival analysis indicates that patients with VM have a poorer clinical outcome in contrast with patients who do not exhibit VM. Figure 2 depicts the related protein fingerprint similarities between ECs and tumor cells displaying VM [57].

## 6. Vasculogenic Mimicry (VM): Functionality of Tumor-Mimicked Vessels in Cancer

What are the biological and functional links between the matrix networks formed by PAS-positive and laminin-positive tumor cells in aggressive melanomas and the normal vasculature lined by endothelium? Before the discovery of VM, several studies on aggressive melanomas (and other tumor types) reported the lining of canals, lakes, and sinuses by tumor cells and their contact with erythrocytes [48,64,65]. However, it is currently unclear whether the vessels formed by tumor cells have any functional relevance for the blood supply to the growing tumor mass and thus can nourish the tumor mass. The prevailing hypothesis is that the erythrocytes found in the extravascular spaces are likely due to leaking blood vessels [9]. The morphological analysis of aggressive melanomas has revealed the presence of networks with PAS-positive patterns, which are associated with poor clinical outcomes [66,67,68,69,70,71,72], and also appeared to converge with blood vessels [4,50]. These networks were found to converge with blood vessels, suggesting anastomosis between the tumor cell-lined networks and the endothelium-lined vasculature. This anastomosis was thought to contribute to the accumulation of erythrocytes in the network scaffold. Therefore, it has been speculated that these tumor cell-lined networks may provide a para-circulation that forms regardless of, or concurrently with, angiogenesis and/or co-option of vessels. Even so, to verify such a complex incident, a more detailed study would be required. An orthotopic model of human uveal melanoma has been developed in immunocompromised mice to further study the production of the unique vascular network pattern distinctive of aggressive melanoma cells [73]. This model also analyses the behavior of liver metastases by uveal melanoma cells (MUM 2B and M619) with the ability to form VM in the target organ [74]. Several studies have demonstrated the existence of a fluid-conducting extracellular matrix meshwork in human cutaneous and uveal melanoma xenograft models, which corresponds to networks exhibiting PAS and positive laminin patterns. These networks are composed of consecutive matrix arcs and loops [75,76]. Different papers used confocal and immune-electron microscopy assays to demonstrate that fluid can be a guide within the endothelial-lined vasculature and extravascular along the channel-like spaces created by PAS-positive and laminin-positive areas. That encloses clusters of tumor cells independently of normal vasculature [77]. Immunohistochemically experiments have shown that this fluid-conducting mesh includes fibrinogen. This denotes the presence of plasma surrounding clusters of tumor cells lined [76]. Further, the erythrocytes that have been noted in many loops and networks of PAS and laminin-positive tumors are likely to be determined from local tumor vessels that are leaky and subject to remodeling. A possible cause of the stabilization of the VM-forming vessels is partly due to the attraction of pericytic cells to the main tumor [78]. PARP inhibitors have been reported to increase mentioned recruitment in VM formation [79,80].

## 7. Vasculogenic Mimicry (VM): Intracellular Signaling

The biological functions of endothelial molecules associated with human tumor cell lines derived from melanoma in the same patient were investigated. This was accomplished by studying gene expression using microarrays in samples obtained from patients with metastatic melanoma [81]. In this microarray, it was observed that the levels of TIE-1 protein (tyrosine kinase receptor-1) and VE-cadherin were highly overexpressed in aggressive tumor cells compared to non-aggressive cells. Moreover, tumor cells did not express CD31 (protein of endothelial cell-specific endothelial cell junction) (Figure 2). The highly aggressive melanoma cells examined in this study were found to express genes that are typically expressed by precursor cells of various cell types, including endothelial, epithelial, pericytes, fibroblasts, hematopoietic, renal, neuronal, and muscle cells, among others. These findings suggest that highly aggressive cells capable of VM formation may undergo a phenotype reversal to an embryonic undifferentiated state. However, the biological significance of this discovery remains to be fully understood. These observations have led to further investigation into the potential relevance of a tumor cell phenotype that challenges our current understanding of how tumor cells can mimic other cell types. Additionally, studies have shown a connection between EphA2 (ephrin type-A receptor 2) and VM. EphA2 is a protein tyrosine kinase that requires ephrin-A1 binding for its phosphorylation and activity (Figure 2); although EphA2 is usually activated by binding with ephrin-A1, it has been reported that in some highly aggressive tumor cells, EphA2 can be constitutively active. Similar to VE-cadherin, the expression of the EphA2 membrane protein was observed only in highly aggressive tumors with the ability to form VM, where it was hyperphosphorylated on tyrosine [82]. When cells were grown on a three-dimensional matrix such as matrigel or collagen and labeled with anti-total phosphotyrosine antibodies, staining revealed that tyrosine phosphorylation was primarily present in areas where tubular network formation occurred. General inhibitors of protein tyrosine kinases, as well as specific EphA2 silencing, decreased the development of vascular networks, indicating a possible and significant role of phosphorylated EphA2 protein in the process of abnormal vessel formation [82]. During VM formation, VE-cadherin and EphA2 are localized to the plasma membrane, specifically in regions of cell-to-cell contact. Inhibiting VE-cadherin led to a rearrangement of EphA2, causing it to move into the cytoplasm or be absent from the cell membrane, and resulted in decreased EphA2 phosphorylation. These results suggest that VE-cadherin may assist in the translocation of EphA2 to the plasma membrane, although the mechanism of this complex relocation remains unknown [83,84]. The PI3K pathway has been shown to have a positive effect on the activity and expression of matrix metalloproteinase 14 (MMP-14) in highly aggressive cells that are capable of forming VM. MMP-14, in turn, activates MMP-2, leading to the cleavage of the 5γ2 laminin chain and the production of γ2 and γ2x fragments. These fragments are then secreted into the extracellular matrix to promote migration in various types of tumors, including breast, colon carcinoma, and hepatocellular carcinoma. In the case of melanoma and ovarian carcinoma, this activation results in the secretion of pro-migratory γ2 and γ2x fragments, leading to VM formation [85,86,87]. The study found that even mildly aggressive melanoma cells, which normally cannot form VM, were able to form vasculogenic-type networks when seeded on collagen gels that had been conditioned by highly aggressive melanoma cells. Although the aggressive cells were eliminated before the apparent formation of tubular networks, examination of cell matrices showed the presence of networks with positive laminin patterns. However, when the matrices were treated with anti-laminin-5γ2 antibodies before seeding the mildly aggressive melanoma cells, they were no longer able to develop tubular networks. These results highlight the significant role of this signaling cascade and laminin-5γ2 in particular. On the other hand, the expression of ADAMTS1 [88] (disintegrin and metalloproteinase with thrombospondin 1 motifs) has been observed to increase in the cell line of fibrosarcoma HT1080132 [89] and correlated with an increase in expression of VE-cadherin, laminin 5γ2, and TIE-1 among others, which led to greater production of staining PAS in a murine xenograft model with the HT1080 cell line, with the consequent capacity for VM formation, due in part to the increase in the aforementioned endothelial proteins. In addition, the deletion of ADAMTS1 through shRNA in the sarcoma cell line of Ewing EW7 decreased the capacity of VM formation in matrigel angiogenesis experiments [89].

In 2001, Mary JC Hendrix [90] and her group published a study that analyzed the functional impact of overexpression of VE-cadherin on aggressive melanoma cells. First, it observed that highly aggressive melanoma cells (C8161, C918, and MUM 2B) expressed high levels of VE-cadherin and TIE-1 in contrast to less aggressive melanoma cells (C81-61, OCM-1A, and MUM-2C respectively). They used HUVEC cells as a positive control for the expression of VE-cadherin, TIE-1, and finally, CD31, the latter of which is normally expressed in ECs. They concluded the study with functional experiments using in vitro 3D angiogenesis development assayed with collagen gel and observed that only highly aggressive melanoma cells C8161 are capable of forming three-dimensional networks in this in vitro 3D model. They also found that both silencing with siRNA technology and blocking VE-cadherin with monoclonal antibodies (2.5 μg/mL) abrogated the VM formation in the three-dimensional model in collagen gels [81,90].

## 8. Vasculogenic Mimicry (VM): Intracellular Signaling, Hypoxia, and Tumor Microenvironments

In the past few decades, the complexity of tumors has been increasingly acknowledged. This has led to a shift in focus in cancer research, where numerous articles are no longer exclusively centered around cancer cells. Instead, researchers are paying more attention to the different components of the tumor microenvironment. One important aspect of the tumor microenvironment is tumor hypoxia, which refers to the low oxygen concentration commonly found in tumors. This condition has been repeatedly associated with malignancy, metastasis, and resistance to therapy in cancer. Many research groups have also linked hypoxia to VM [91,92,93,94]. HIF promotes stemness and differentiation potential in highly malignant tumor cells, transforming them into more mobile cells through the Epithelial–Endothelial Transition (EET) process induced by hypoxia. This results in the upregulation of transcription factors Twist and Snail, leading to the downregulation of tight junction protein E-cadherin and upregulation of molecules related to angiogenesis, such as VE-cadherin and fibronectin. These cancer stem cells display characteristics of ECs, forming a VM network through intracellular signaling pathways and remodeling of the Extracellular Matrix (ECM). This network provides a space for tumor cell migration and the formation of duct-like structures that extend into the vascular network, transporting nutrients to the tumor cells [95]. In addition, HIF-1 has been shown to directly regulate the expression of several molecules associated with VM, including VEGF, Twist, LOX, and MMP2 [93].

Studies have shown that hypoxia promotes VM through different signaling pathways. For example, reactive oxygen species (ROS)-mediated stabilization of HIF1α activates the met proto-oncogene, inducing tube formation on matrigel in melanoma cells [96]. Moreover, HIF1α and HIF2α have been shown to promote in vitro tube formation by upregulating vascular endothelial growth factors (VEGF) A, C, and D, as well as VEGF receptor (VEGFR1-2) expression [97]. In a recent study, NRP-1 was also found to be involved in this process in lung adenocarcinoma. Moreover, the study indicates that MMP2, VE-cadherin, and vimentin are impacted by the mechanisms involved in lung adenocarcinoma cell metastasis and VM formation, with HIF-1α playing a crucial role in this process by upregulating the expression of NRP1. The study suggests that targeting NRP1 could be a valuable therapeutic strategy to prevent lung adenocarcinoma metastasis and progression [98]. In triple-negative breast cancer, hypoxia increases the subpopulation of CD133+ cells, commonly regarded as cancer stem cells, through a Twist1-mediated mechanism. This population shift seems to enhance tube formation, as CD133+ cells are found to line VM-like tubes [99]. Furthermore, HIF1α can promote tube formation in hepatocellular carcinoma by upregulating lysyl oxidase-like 2 (LOXL2), which is involved in collagen cross-linkage during extracellular matrix (ECM) remodeling [100]. The extracellular matrix (ECM) itself can also play a fundamental role in regulating VM. For example, the NC11 domain in collagen XVI can trigger tube formation in oral squamous cell carcinoma by inducing VEGFR1/2 expression [101]. In contrast, the presence of collagen I alter the vascular potential of pancreatic ductal adenocarcinoma (PDAC) CSCs, decreasing the secretion of pro-angiogenic factors and the expression of VEGFR2, thus hindering VM formation in PDAC [102]. Furthermore, ECM architecture can influence VM; for instance, collagen matrices with small pores and short fibers induce integrin-β1 expression and hence VM. Furthermore, integrin-β1 KO HT1080 cells decrease the VM formation in matrigel [103,104].

The importance of non-cancer cells within the tumor stroma is also gaining attention in the study of VM. Tumor-Associated Macrophages (TAMs) seem to promote VM formation in glioblastoma multiforme by increasing the expression of cyclooxygenase 2 in tumor cells [105]. Cancer-Associated Fibroblasts (CAFs) can also be determinants in VM formation. In a recent study, vasculogenic murine melanoma cells were injected into mice carrying a CAF-specific deletion for the matricellular protein CCN2. As a result, the absence of fibroblast-derived CCN2 reduced tumor vasculature, including VM [106]. Finally, a recent publication by Thijssen et al. [78] showed that PAS+ tissues in human cutaneous melanoma stained positive for pericyte marker α-smooth muscle actin (αSMA) within the ECM networks lined by tumor cells. Furthermore, when VM+ tumor cells were co-cultured with pericytes, there was a stabilization of the VM networks for up to 96 h. Pericyte recruitment to VM networks was shown to be dependent on PDBF-B signaling, whereas the addition of STI-571 (imatinib mesylate) to inhibit PDGF receptor hindered VM as well as tumor growth.

## 9. Vasculogenic Mimicry (VM): Intracellular Signaling, Focus on Non-Endothelial VE-Cadherin

The endothelium plays a crucial role in regulating vascular permeability by utilizing proteins, plasma, and circulating cells. The specific function of controlling cell permeability is governed by both transcellular and paracellular pathways [107,108]. The transcellular pathway is mediated by complex vesicular systems that contain receptors for circulating proteins. These vesicles arranged on the endothelial side apical transport plasma components to the basal face of the endothelial membrane and into the sub-endothelial space. Para-cellular permeability is controlled by specific adhesion molecules present at cell–cell junctions and by the ability of ECs to retract and eventually open cell–cell junctions [109]. Stimulation by soluble agents such as histamine or VEGF increases vascular permeability and acts on both systems, increasing the number and organization of vesicles and decreasing the strength of inter-endothelial adhesion [110,111,112]. VE-cadherin is a transmembrane protein commonly expressed in the endothelium, where it is responsible for cell–cell adhesion [113], and in vivo, experimental models have shown that VE-cadherin-deficient mice die from severe vascular defects [114]. Although VE-cadherin used to be specific in normal ECs, its aggressive and VM formation has been required in melanoma. Paradoxically, VE-cadherin can be found in highly aggressive tumor cells but not mildly aggressive ones. Furthermore, the absence of its expression in melanoma implied the loss of the formation of VM [90].

One study, which was published in 2014, has revealed that the Y658 residue of VE-cadherin can be phosphorylated by FAK in in vitro and in vivo models of ECs associated with tumor cells, thus identifying that FAK plays a pivotal role in the dynamics of cell permeabilization in the metastatic tumor endothelium [115]. On the other hand, the phosphorylation of VE-cadherin Y685 is mediated by Src, having an important role in vascular permeability in vivo [116,117]. Finally, the phosphorylation of Y731 has been observed to play an important role in the internalization of VE-cadherin through the phenomenon called diapedesis dependent on histamine stimulation that causes the extravasation of neutrophils and leukocytes on the vascular endothelium [117], on the other hand, Y731F-mutated mice were affected in neutrophil extravasation over normal vascular endothelium. Our group recently reported the important role of Y658 of VE-cadherin in the formation of VM; indeed, FAKi provoked the VE-cadherin/p120/Kaiso decoupling on the nucleus enhancing the transcription Kaiso activity upon Kaiso-dependent genes expression (Cyclin D1 or CCND1 and WNT11) [118] (Figure 2 and Figure 3). PTPRB (receptor-type tyrosine-protein phosphatase beta) or VE-PTP, called VE-cadherin-dependent endothelial context protein [119,120], is intrinsically expressed and shown in ECs. Still, later the expression of VE-PTP has been demonstrated in VM cells of origin uveal melanoma. Moreover, silencing VE-PTP produces a decoupling of the VE-cadherin/p120 complex and produces more VE-cadherin degradation by autophagy, enhancing the inhibition of VM formation by tumor cells [121]. Following in the VM production by tumor cells, recently, our group reported that FAK plays an essential role in the acquisition of VM property, permitting the union of VE-cadherin to β-catenin/TCF-4 (Transcription factor-4) to enhance the transcription activity from TCF-4 dependent genes (c-Myc, Twsit-1, and S1PR1) [122].

## 10. Concluding Remarks

Vessel formation (independent of the original form) has produced an exciting point for the scientific community in the last decade, recently following this enigmatic paradigm in tumor vessel formation, several seminal articles [123,124,125] put the focus on the individualized study of the single-cell analysis approach. This review presents a specific strategy and insight that tumor cells use to escape the adverse microenvironment through transdifferentiation, leading to a “cartoon” of vascular endothelial-like cells. VE-cadherin, along with FAK activity and tethered β-catenin, acts as sensors of the tumor microenvironment. Under unfavorable conditions, they enable the formation of VM to allow cancer cells to attract nutrients and oxygen. This transdifferentiation allows the tumor mass to develop an irrigated avascular network, which helps to avoid nutrient and oxygen shortages within the tumor by creating para-vascular pathways parallel to normal vessel formation. Oxygen depletion in growing tumors often leads to hypoxia and the adaptation of tumor cells to a changing microenvironment. Hypoxia-inducible factors (HIFs) and hypoxia-responsive elements (HRE) play a crucial role in this process by inducing genes for tumor cell adaptation. Hypoxia has been reported to promote VM in a wide variety of tumor cell lines through the upregulation of genes such as VEGF-A, VEGFR-1, EphA2, Twist, Nodal, COX-2, and VE-cadherin [93]. The expression of HIF-1α and HIF-2α is important in regulating the expression of VE-cadherin and modulating VM in various cancer types. Hypoxia can also influence VM through BNIP3 and mTOR signaling pathways [96]. Therefore, targeting HIFs, HREs, and other hypoxia-induced signaling pathways may offer new therapeutic strategies for inhibiting VM and reducing tumor malignancy. As reviewed, the deregulation of endothelial markers expression by the most aggressive tumor cells, as well as changes in cell phenotype, play a crucial role in aggressive metastatic behavior. These changes have functional and translational significance. The molecular pathways involved in VM formation have revealed that VE-cadherin is a key component and should therefore be considered in the development of novel treatment strategies that target tumor cell plasticity and metastatic properties. This is particularly important as it is related to disease recurrence and drug resistance, which pose significant challenges for future research [126,127,128]. New concepts are needed to discern the specific signaling distinctiveness of VE-cadherin (apart from its role in the context of the endothelial cell [39]) that affects tumor cell biology and is related to hostile environment from VM development. Special attention should be paid to the identification of tumor cell precursors committed to the acquisition of the VM phenotype/VE-cadherin expression and the interactome linking VE-cadherin to cell trans-differentiation. One of the precursors may have the latency or dormancy of aggressive tumor cells waiting for the best moment to colonize pre-metastatic niches as far away from the primary tumor, thus opting for the best survival conditions to carry out metastasis [129,130,131]. The elevated levels of nuclear VE-cadherin and p-VE-cadherin Y658 in VM-capable cells are critical to understanding the specific role of non-vascular VE-cadherin in acquiring invasive properties. Our ability to effectively eliminate cancer is limited due to the heterogeneous subpopulations that contribute to tumor formation, including the diverse development of the vasculature. Hypoxia induced by rapid tumor growth or conventional therapies can also catalyze and activate VM formation, leading to unintended consequences. To overcome these limitations, a deeper understanding of VM biology is necessary. Therefore, it may be prudent to consider using new agents that target VM pathways [118] associated with vessel co-option formation [132], such as integrin [43] and other adhesion molecules, in combination with “classical” anti-angiogenic agents [5,133].

## 11. Open Questions

Further understanding of the distinct signaling functions of VE-cadherin outside of its role in ECs is necessary to comprehend its impact on tumor cell biology and its association with hostile/VM development.

Special attention should be paid to the identification of tumor cell precursors committed to the acquisition of the VM phenotype/VE-cadherin expression and the interactome linking VE-cadherin to cell trans-differentiation.

Consider the application of new therapeutic agents to target VM pathways related to vessel co-option formation (integrin and other adhesion molecules) with “classical” anti-angiogenic agents.

## Figures and Tables

**Figure 1 ijms-24-09343-f001:**
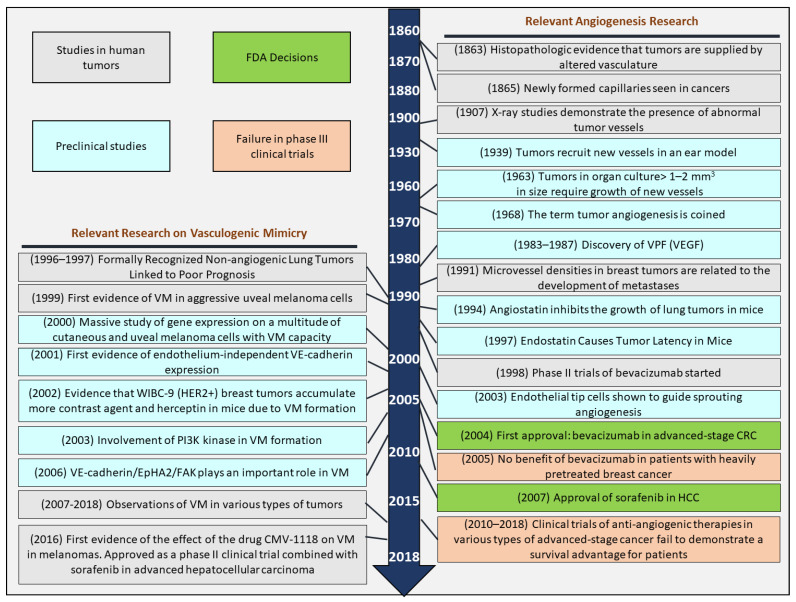
A glance at angiogenesis past. Historical timeline of selected relevant investigations in VM and angiogenesis. Pre-clinical experimental studies (Blue Box), histopathology reports of human tumors (Brown Box), clinical trial results (Red Box), FDA decisions (green box), and other events are key from the middle of the 19th century to the present day. Research findings related to tumor angiogenesis are shown on the right, and those related to VM are shown on the left.

**Figure 2 ijms-24-09343-f002:**
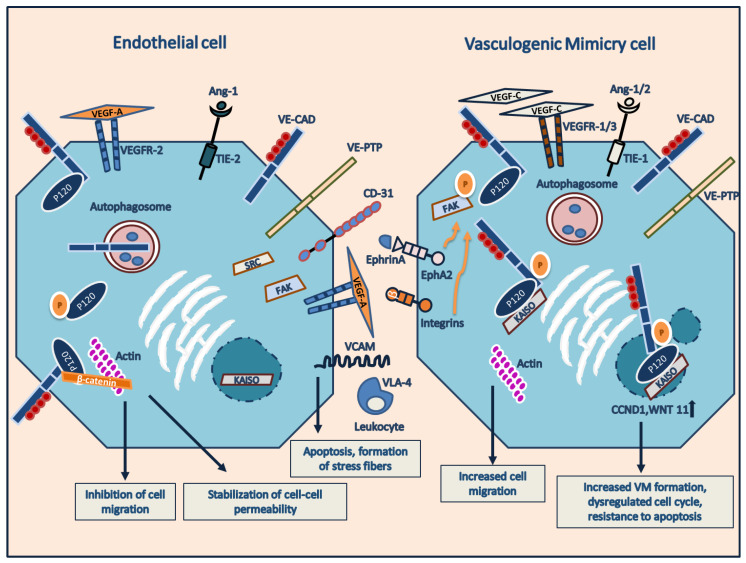
Mechanism of vascularization in cancer, VM. Schematic representation of protein expression compared to ECs versus aggressive tumor cells that produce the VM phenomenon [58,59,60,61,62,63]. More detail is provided in the text.

**Figure 3 ijms-24-09343-f003:**
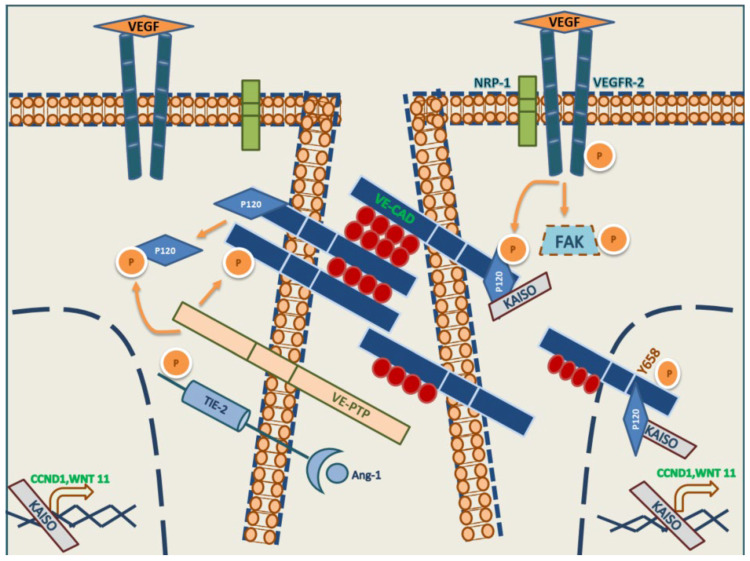
Mechanism of vascularization in cancer, VM. Intracellular signaling of VE-cadherin in VM cells. More detail is provided in the text.

## Data Availability

Not applicable.

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
