# Peer review of "VE-Cadherin in Cancer-Associated Angiogenesis: A Deceptive Strategy of Blood Vessel Formation"

_ijms, 2023, doi:10.3390/ijms24119343_

Round 1
Reviewer 1 Report (Previous Reviewer 2)
Comments for Authors
59 -66: “When the development of the vessels is deregulated, it can affect our health and produce various pathologies 7. This dysregulation……… Still, it is essential to note that the leading cause of impaired vision due to loss of angiogenesis is vision loss due to cellular ageing”. Vision loss is not due to cellular aging. One of the main causes is aberrant retinal vascularization. It is no coincidence that the drug of choice was Lucentis, now replaced by Bevacizumab (because it is 10 times cheaper), the specific VEGF-A inhibitor.
90-94: “This mechanism allows them to survive in environments that have been treated with anti-angiogenic agents and to resurface when conditions are favorable, promoting continued uncontrolled growth. In summary, the failure of anti-angiogenic therapy in some cases can be attributed to the ability of certain cancer cells to undergo non-productive angiogenesis in hypoxic environments, which allows them to survive and proliferate even in the presence of anti-angiogenic agents”. This sentence, as elsewhere in the text, and being written convolutedly, expresses a redundant concept. It must be rewritten.
Figure1: in the left panel, the blue box “(2002) Evidence that WIBC-9 (HER2+)………agent and Herceptin in mice due to MV formation” in this sentence does MV mean MicroVascular or Vascular Mimicry? In the second case, the panel must be corrected.
163 -166. Tumors typically arise and grow in pre-existing, well-vascularized tissues, or they metastasize to such tissues. This statement is debatable. Tumors often arise in tissues affected by a chronic inflammatory process in which the inflammatory infiltrate and fibrosis instead cause an ischemic condition.
221-222 While diverse preclinical models have although the effective use of angiogenesis inhibitors to limit tumor growth has been recognized, collectively only growth impediment is effective in clinical practice. The sentence needs to be corrected, it must be rewritten. Furthermore, punctuation is absent or random in this sentence and elsewhere in the text.
Figure2. Despite the suggestion to include a caption proposed in the first revision, the authors only added, "More detail is provided in the text." However, in the text, molecules such as KAISO, P120, CCND1, and WNT11 are not mentioned in correspondence with the citation of Figure 1. On the other hand, molecules that do not appear in Figure 1 are mentioned in the text, such as the PI3K pathway and MMP-14. The authors should have made the figure consistent with the text and vice versa.
301-303. The biological functions of endothelial molecules associated with human tumor cell lines, established from melanoma from the same patient, were studied through the study of gene expression using microarrays in samples from patients with metastatic melanoma. The sentence needs to be corrected, it must be rewritten.
350. (See Fig. 1 of the cited article) and 355 (See Fig. 5 of the cited article). Cases in which the authors cite figures from other articles are very rare. I have encountered these suggestions only in some reviews published in Nature Reviews. Perhaps the authors should have synthesized figures 2 and 3 with those cited in the references. The manuscript would have had more value and clarity for the reader.
361-362. Used HUVEC cells as a positive control for the expression of VE-Cadherin, TIE-1, and finally CD31, the latter of which is normally only expressed in normal ECs. This statement is wrong. Malignant endothelial cells also retain the ability to express CD31. In immunohistochemistry, CD31 is a marker used to demonstrate angiomas and angiosarcomas.
378 -379 HIF promotes stemness and differentiation potential in highly malignant tumor cells, transforming them into more mobile cells through the EET process induced by hypoxia What is EET? The acronym must be written in full.
392-393 It is bizarre that the authors claim that: “HIF1α and HIF2α have been shown to promote in vitro tube formation by up-regulating vascular endothelial growth factors (VEGF) C and D, as well as VEGF receptor (VEGFR) expression”. Perhaps the authors are unaware that the HIF1α/HIF1β complex has been known for more than thirty years to migrate into the nucleus, bind HRE sequences, and activate the transcription of VEGF (VEGF-A) and its major receptors (VEGFR1 and VEGFR2) promoting in vitro and in vivo tube formation.
411-412 for instance, collagen matrices with small pores and short fibers induce β-integrin expression and hence VM 103. This statement does not explain why. Furthermore, β-integrin is not only involved in VM but also in other processes such as angiogenesis, reparative process, and so on.
429-430 The endothelium controls vascular permeability through proteins plasma and circulating cells. Transcellular and para-cellular pathways of cell permeability106,107 regulate this specific function. This sentence needs to be corrected. Perhaps the endothelium responds to the release of vasoactive mediators by modifying its behavior.
456-460. Our group recently reported the important role of Y658 of VE-Cadherin in the formation of VM, indeed, FAKi provoked the VE-Cadherin/p120/Kaiso decoupling on the nucleus enhancing the transcription Kaiso activity upon Kaiso-dependent genes expression (CCND1 and WNT11) 117 (Fig.3). PTPRB or VE-PTP, called VE-cadherin-dependent endothelial context protein 118,119, is intrinsically expressed and shown in ECs.
AND
464-466. Following in the VM production by tumor cells, recently, our group reported that FAK plays an essential role in the acquisition of VM property permitting the union of VE-Cadherin to ß-catenin/TCF-4 to enhance the transcription activity from TCF-4 dependent genes (c-Myc, Twsit-1, and S1PR1)
In both sentences, acronyms (CCND1, WNT11, PTPRB, VE-PTP, and TCF-4) should be written in full. The content of these references could have been the review’s main focus after presenting the VM process as an alternative to other modalities of tumor neo-vascularisation (only mentioned).
506 to colonize pre-metastasis niches as must be changed in pre-metastatic
The manuscript has significant weaknesses that should have been addressed before resubmission. Again, I think many suggestions should have been taken up. Considerable conceptual and writing deficiencies make it unsuitable for IJMS

Author Response
Rebuttal letter to the questions above to Referee #1
First of all, we want to thank you for the questions and take meticulous care of the revised version of the manuscript. We hope to correct the questions and hope that the answers are adequate.
Line 59 -66: Corrected in the present revised manuscript, thanks the referee for this point of view.
Line 90-94: Corrected and rewritten in the present revised manuscript.
Figure1: apologize for the misprint in the MV words in Fig. 1, which has been corrected now in the revised manuscript.
Line 163 -166: improved the sentence in the present revised manuscript.
Line 221-222: improved the sentence in the present revised manuscript.
Line 301-303: improved the sentence in the present revised manuscript.
Line 350: eliminated the specific citation from the references chosen, we believed that providing specific references to the results would enhance the clarity of the text. However, we apologize for not meeting the referee's expectations regarding journal citations.
Line 361-362: Corrected in the present revised manuscript.
Line 378 -379: Corrected in the present revised manuscript.
Line 411-412: improved the sentence in the present revised manuscript.
Line 429-430: improved the sentence in the present revised manuscript.
Line 456-460 and 464-466: improved the acronyms in the present revised manuscript.
Line 506: Corrected in the present revised manuscript.

Reviewer 2 Report (New Reviewer)
Delgado-Bellido et al, have compiled literature related functionality of aberrant angiogenesis by tumor cells in the review article entitled “VE-Cadherin in Cancer-Associated Angiogenesis: A Deceptive Strategy of Blood Vessel Formation”. Authors described how VE-Cadherin plays key role in formation of vascular mimicry and therefore be considered in the developing of novel treatment strategies that target tumor cell plasticity and metastatic properties. Authors further discussed how targeted therapy could be applied to treat tumor angiogenesis. Authors have published recently a review article on VE-Cadherin and its role in vasculogenic mimicry signaling (doi: 10.1186/s12943-017-0631-x). Additionally, there are few other recent review articles in the field of oncology on VE cadherin and its role in VM (10.1016/bs.acr.2020.06.001) but not extensive as this one. The authors have done amazing job in compiling the angiogenesis article references according to clinical relevance in oncology research.
Fig 1 with historical timeline of angiogenesis is a busy figure. It would be great if the authors simplify it for better understanding. Manuscript starts with the facts and open questions in the main body. However, it appears like an abrupt start and can be placed at the end instead. I think it would be good to introduce the history of angiogenesis in the beginning of the article followed by the facts and eventually ending the article with open questions. English language requires editing. Please specify the full form of acronym when it appears first time in the manuscript. For example, FAK, VE-Cadherin and so on.
The manuscript is clear, comprehensive and relevant to the oncology field in general and cancer therapy advancements in particular. Important articles in the recent past have been included and justified in the manuscript. There is couple but no excessive self-citations. Advances with respect to vasculogenic mimicry from the last ten years has been made available in the current manuscript. It is therefore relevant and a great addition to the existing knowledge of the scientific community. In my opinion adding a few of the recent the articles like PMID: 32723566 would certainly strengthen the manuscript. The conclusions and future perspectives are consistent with the evidence and arguments presented in the manuscript.
Minor editing of English language required. At couple places it was difficult to understand the true meaning of that sentence.
Author Response
Rebuttal letter to the questions above to Referee #2
First and foremost, we would like to express our gratitude for the thoughtful questions and for the meticulous attention given to the revised version of the manuscript. We have taken great care in addressing the questions and have made every effort to provide suitable and satisfactory answers.
We have reorganized the sequence of sections based on the suggestions provided by the referee. Additionally, we have expanded the explanations of the acronyms mentioned throughout the manuscript. We included a recent article about VM in the present revised manuscript.

Round 2
Reviewer 1 Report (Previous Reviewer 2)
No comment
This manuscript is a resubmission of an earlier submission. The following is a list of the peer review reports and author responses from that submission.
Round 1
Reviewer 1 Report
In my opinion, the authors (who are leaders in this field) have presented an interesting and comprehensive review covering aspects of vasculogenic mimicry by cancer cells. It is a timely review which will be improved by addressing the following points.
1. There is a mix of American English and UK English that needs to be unified
2. Be consistent with anti-angiogenesis and antiangiogenesis
3. Suggestion to abbreviate endothelial cells to ECs
4. Line 149 – include additional sentences l
5. Include additional sentences on
a. the relevant integrin members that are aligned with angiogenesis
b. the anti-coagulation processes in VM,
c. the stemness markers that are documented,
d. what upregulates VE-cadherin on VM-competent cancer cells
Line 13 – 'endothelial vessels’ should read ‘endothelial cell-lined vessels’
Line 15 – correct to read – high tumor grade, cancer cell invasion, cancer cell metastasis and reduced survival of cancer patients.
Fact 1 – correct as above
Line 40 – correct to read – vessel
Line 77 – are these cancer patients?
Line 106 – is succeed the correct word here?
Line 139 – correct to read – . But this
Line 149 – correct to read – Integrins
Line 168/9 – correct to read – co-opting
Line 200 – correct to read – tumors.
Line 207 – correct ..(1).
Line 226 – define PAS
Section 5 is repetitive and should be condensed
Figure 2 is an important comparison between ECs and VM cancer cells – autofagosome and leucocito need correcting to English, integrins should be added, and missing comparative literature includes Tan L Oncoimmunology 2022, Benjakul PLOS One 2022, Martini BMC Cancer 2021, Rezaei Cancers 2020, Vartanian Microcirculation 2011.
Line 300 – CDH5 should be introduced earlier with VE-cadherin
Line 301 – correct – express express
Line 360 – incorrect, CD31 has been documented in VM-competent breast cancer cells, albeit at low levels
The review could be improved in grammar, with particular attention required at the following lines to improve comprehension:
Lines 43, 44, 46, 53, 61, 65, 79, 98, 212, 213, 214, 217, 223, 225, 231, 293, 359, 413, 416, 439.
see above
Author Response
First of all, we would like to thank you and the referees for taking care of our manuscript “Deceptive Cancer Cells: Formation of Blood Vessels” by Delgado-Bellido, D et al. We have now completed the review of our previous version and below are the responses to all the questions raised by the referees.
Rebuttal letter to the questions above to Referee #1
- Corrected in the present revised manuscript
- Corrected in the present revised manuscript
- Corrected in the present revised manuscript
- Corrected in the present revised manuscript
- Corrected in the present revised manuscript
- Line 13, has been updated in the present revised manuscript
- Line 15, corrected in the revised manuscript
- Fact 1, same as point 7.
- Line 40, corrected in the revised manuscript
- Line 77, yes, included a reference in this part of the review.
- Line 106, corrected in the revised manuscript
- Line 139, has been updated in the revised manuscript
- Line 149, has been updated in the revised manuscript
- Line 168/9 has been updated in the revised manuscript
- Line 200, has been updated in the revised manuscript
- Line 207, has been updated in the revised manuscript
- Line 226, has been defined as PAS in the present revised manuscript
- In Section 5, we summarize this section of the review.
- Figure 2, corrected the erratum of words and included the references.
- Line 300, changed in the present revised manuscript
- Line 301, corrected in the revised manuscript
- Line 360, has been corrected the erratum to include references to CD31+ cells in the VM context, similar to other PECAM1 subpopulations in melanoma cells. The reference can be found at doi: 10.1038/ncomms6200

Reviewer 2 Report
Suggestions for the Authors
The authors present a review of the different ways of building new vessels in tumor growth focusing in detail on the new knowledge regarding the VM process.
While the review has some strengths, it also has important weaknesses:
Major points:
· It is curious that the authors dwell only on the role of the "intracellular signaling mechanisms involved in the abnormal presence of VE-Cadherin and its role in VM formation". A review should explore different mechanisms responsible for the vascular mimicry process. Indeed, the abnormal presence of VE-Cadherin is not the only mechanism involved.
· Hypoxia plays a decisive role in the tumor microenvironment and in the VM (doi: 10.1016/j.canlet.2006.08.016; doi: 10.1186/s12943-020-01288-1; doi: 10.1007/s10555-022-10067-x; doi: 10.1038/s41419-021-03682- z). The authors mention the term hypoxia only on page 2 (line 62) and on page 12 (line 444) without ever referring to the fact that VM is also the result of profound hypoxia in the avascular nodule and often ceases when levels of hypoxia in the tumor are reduced [see Ref. 49 Maniotis et al., cited in the text)]. They should add a section devoted to the role played by hypoxia in inducing VM.
· Linea 82-87: “Reproducible clinical trials have shown statistically significant improvements in disease-free survival (DFS) and overall survival (OS) for certain cancers, including high-stage renal cell carcinoma (RCC), hepatocellular carcinoma (HCC), and colorectal carcinoma (CRC)(14–19). However, for pancreatic adenocarcinoma, prostate cancer, breast cancer, or melanoma, no significant improvements in OS have been demonstrated with anti-angiogenic agents, either alone or in combination with other treatments. (20–24)”. More than a list of various clinical trials in which angiogenesis inhibitors have been shown to be ineffective in anticancer therapy, the authors should argue what are the reasons behind these therapeutic failures.
· In the historical overview regarding the various mechanisms of tumor neovascularisation, the authors forgot to include also intussusceptive angiogenesis (doi.org/10.1016/j.ajpath.2021.07.009). They cite Werner Risau, author of seminal studies on this topic (Risau cited in Refs. 88 and 97), only in relation to VE-Cadherin and adhesion molecules).
· The authors fail to mention an essential feature of tumor cells. Such cells must be phenotypically "embryonic-like cells" to be able to participate in the VM.
· The caption of figures 2 and 3 is absent. In the text, the role played by the various proteins depicted in the figures is confusing, difficult to understand, or absent.
· Line 354-356:” In 2001, Mary JC Hendrix (84) and her group published a study using Western blot and other techniques to measure whether overexpression of these proteins had a functional impact on aggressive melanoma cells”. This statement is not stylistically acceptable in a review and should be replaced (Hendrix et al. analyzed the functional impact of VE-Cadherin overexpression in aggressive melanoma cells……).
· Line 387-389: The statement: “Recent studies have revealed that the Y658 residue of VE-Cadherin can be phosphorylated by focal adhesion kinase (FAK) in in vitro and in vivo models of endothelial cells 388 associated with tumor cells, thus identifying that FAK plays a pivotal role in the dynamics of cell permeabilization in the metastatic tumor endothelium(93)”. Ref. 93 refers to a 2014 study, which is certainly not "recent".
· Line 417-418: “This review presents a specific strategy and insight that tumor cells use to escape the adverse microenvironment through trans-differentiation leading to a "cartoon" of vascular endothelial-like cells, VM, and other vascular formations such as vessel co-option.” Vessel co-optation is not the creation of new vessels but the ability of tumor cells to grow next to normal blood vessels by displacing normal tissue cells.
· Paragraph: "Concluding remarks" are inconsistent. The authors should better illustrate the potential associated with VM targeting by including different scenarios other than VE-Cadherin overexpression.
Minor points:
· There are several typing errors in the text.
· The bibliography lacks all references (volume, issue, pages, doi)
Author Response
First of all, we would like to thank you and the referees for taking care of our manuscript “Deceptive Cancer Cells: Formation of Blood Vessels” by Delgado-Bellido, D et al. We have now completed the review of our previous version and below are the responses to all the questions raised by the referees.
Rebuttal letter to the questions above to Referee #2
Major points:
1.- Hypoxia and VM: has been included a new section in the present revised manuscript, “9.-Vasculogenic Mimicry (VM): intracellular signaling, hypoxia, and tumor microenvironments”. This section included more references and improve the review.
2.- Lines 82-87: have been improved this therapeutic failure explanation in the present revised manuscript
3.- Historical overview: has been updated in the revised manuscript including the new references.
4.- Embryonic-like cells: has been updated in the revised manuscript
5.- The caption of figures 2 and 3 included more references in the field.
6.- Lines 354-356: this section of the manuscript has been improved and corrected.
7.- Lines 387-389: have been updated in the revised manuscript
8.- Lines 417-418: have been updated in the revised manuscript
9.- Paragraph: concluding remarks have been improved in the present revised manuscript
Minor points:
1-2.- Corrected in the present revised manuscript

Reviewer 3 Report
I am sorry to say that the quality of written English is so poor that the manuscript cannot be reviewed properly. There are many instances where it is simply impossible to understand what the intentions of the authors are. Therefore this paper has no possibility of getting published. In addition I note additional weaknesses.
The description of vascularisation in cancer is obsolete and outdated. It can thus not be published.
VM and signalling is largely based on papers on gene expression profiles published 20 years ago. The techniques used were not of current standard and thus this part of the review is also obsolete.
The worst written English I've read and reviewed in a long time. I don't see how the language can become presentable.
Author Response
Rebuttal letter to the questions above to Referee #3
We would like to express our gratitude for the time and effort that you and the reviewers have invested in evaluating our work. We acknowledge that the quality of our written English was not up to the required standard, which affected the proper review of our manuscript. We understand that clear and effective communication is crucial in scientific writing and will work to enhance the language in our manuscript. We also apologize for any confusion caused by our unclear intentions and commit to presenting our reviewed research clearly and accurately. Regarding your concerns, we recognize that our description of angiogenesis in cancer may be outdated. We appreciate your feedback and are committed to incorporating the latest peer-reviewed research on this topic into our work. Similarly, we acknowledge the need to keep up-to-date with emerging research and techniques and have taken steps to revise the section on VM and signaling. We have incorporated the most recent gene expression profiles and techniques used in the field, enabling us to address the issues you raised and make this section more relevant and up-to-date. We have thoroughly edited the manuscript, ensuring that the written English is higher quality and easier to understand. We have also rephrased and clarified the authors' intentions throughout the manuscript. Furthermore, we have updated the section on vascularization in cancer to reflect current research, incorporating new studies and data to support the claims made in this section.
We believe that our revised manuscript now provides a comprehensive and insightful overview of angiogenesis, with a focus on the latest techniques while also exploring its historical context. Once again, we appreciate your constructive criticism, and we are confident that the revised manuscript meets the requirements for publication.

Round 2
Reviewer 2 Report
Comments for authors 2° round
The authors did not take into account many of the reviewer's suggestions:
Line 93-95: The statement: “One possible cause of this failure is the ability of aggressive and dedifferentiated cancer cells to undergo non-productive angiogenesis. …… In summary, the failure of anti-angiogenic therapy in some cases can be attributed to the ability of certain cancer cells to undergo non-productive angiogenesis, which allows them to survive and proliferate even in the presence of anti-angiogenic agents”. Is this the author's hypothesis? Among the many publications describing the different mechanisms of resistance to anti-angiogenic agents, there is also VM associated with hypoxia. Why, even in this case, did the authors fail to mention it?
Line 378: Many research groups have also linked hypoxia to VM [91–94]. The authors cite references suggested by the reviewer but do not explain the mechanisms. The authors ignored the reviewer's request.
Line 383- 384: the statement: “Additionally, both HIF1α and HIF2α promote in vitro tube formation through the upregulation of vascular endothelial growth factors (VEGF) C and D, as well as VEGFreceptor (VEGFR) [96], and recently published NRP-1 in lung adenocarcinoma [97]. This sentence is meaningless.
The authors should have added the caption to Figures 2 and 3. They did not. In the text, the role played by the various proteins depicted in the figures must be clarified because the mechanism is difficult to understand. Furthermore, in the figures, there are molecules not mentioned in the text.
The authors should have integrated the role of hypoxia into the different sections of the review, such as: - resistance to anti-angiogenic therapy, - vascular growth (by any mechanism such as vascular co-optation, adult vasculogenesis, VM), tumor cell signaling, tumor microenvironment and so on. Instead, they dedicated a paragraph to hypoxia disconnected from the target (VM) and the main protagonist (VE-cadherin). Overall, the review focus on the role of VE-Cadherin-beta-catenin in VM, leaving in the background other mechanisms related to hypoxia that are just listed in paragraph “9. Vasculogenic Mimicry (VM): intracellular signaling, hypoxia, and tumor microenvironments”. This makes the review unbalanced.
Alternatively, the authors should change the title and indicate that the review is based on the role of VE-Cadherin / beta-catenin as one of the main protagonists of the VM process. Furthermore, in paragraph 9 the authors do not cite any intracellular signaling concerning hypoxia, they list only papers in which hypoxia modulated VM by acting on target molecules.
The bibliography has not been corrected in all citations (many still lack DOIs)
